# Relationship between Psychological Distress and Continuous Sedentary Behavior in Healthy Older Adults

**DOI:** 10.3390/medicina55070324

**Published:** 2019-06-30

**Authors:** Yutaka Owari, Nobuyuki Miyatake

**Affiliations:** 1Shikoku Medical College, Utazu, Kagawa 769-0205, Japan; 2Department of Hygiene, Faculty of Medicine, Kagawa University, Miki, Kagawa 761-0793, Japan

**Keywords:** healthy elderly people, psychological distress, sedentary behaviors, structural equation modeling (SEM)

## Abstract

*Background*: Our purpose is to clarify whether psychological distress (PD) affects the rate of continuous sedentary behavior (CSB). *Materials and Methods*: In this secondary analysis, a sample population of 80 healthy older adults aged 65 years or older participated in a health club of college A from 2016 to 2017. We conducted Structural Equation Modeling (SEM) using the cross-lagged and synchronous effects models. We adopted the following as proxy variables: CSB (based on the ratio of 1.5 METs sessions or more continuing for over 30 min) CSB and PD (based on the Kessler psychological distress scale: K6). *Results*: “2016 K6” had a significant influence on “2017 CSB” (standardization factor (β) = 0.136, *p* = 0.020) using the cross-lagged effects model, and “2017 K6” significantly influenced “2017 CSB” (β = 0.166, *p* = 0.039) using the synchronous effects model. Fit indices were Adjusted Goodness-of-Fit Index (AGFI) = 0.990, Confirmatory Fit Index (CFI) = 1.000, and Root Mean Square Error of Approximation (RMSEA) = 0.000. *Conclusion*: The results suggest that PD may affect the ratio of CSB one year later.

## 1. Introduction

Japan has one of the largest aging populations in the world. According to the Cabinet Office Government of Japan, Japan’s aging rate (the percentage of the population over 65 years old) was 27.3% in 2016 [1]. Therefore, maintaining the health of the older adults’ population is important, and solving this problem would also be meaningful to other countries. The World Health Organization (WHO) stated that “health is a state of complete physical, mental and social well-being and not merely the absence of disease or infirmity” (WHO, 1946 [2]). It is well known that there is a strong correlation between physical and mental factors.

Many studies have shown the relationship between physical and mental factors in patients with depression. Although some studies reported that a large amount of physical activity reduced the likelihood of depression [3,4,5,6,7], others showed the contrary [8,9]. Furthermore, some studies showed that interrupting sedentary behaviors was effective in improving the quality of life [10,11], while others reported a trade-off between the time spent on exercise and sitting [12].

Based on the above, we hypothesized that physical factors were caused by mental factors. More specifically, we assumed that improvements in psychological distress (PD, mental factor) caused the decrease in the rate of continuous sedentary behavior (CSB, physical factor). Many studies have shown that a reduction in sitting time leads to reduced health risk [11,12]. However, recent studies have reported that even with the same total sitting time, there were differences in health risks depending on whether or not the sedentary behaviors were interrupted [10,13]. Therefore, in this study, the causal relationship between psychological distress and CSB were clarified.

## 2. Materials and Methods

### 2.1. Study Design

In this second analysis from our previous reports [14,15], we conducted a longitudinal study using a “two-wave” panel data. We adopted the following as proxy variables: CSB and PD that are based on the rate of 1.5 Metabolic equivalents (METs) or more that continued for over 30 min and the K6 scores, respectively. To define our purpose, we adopted the following hypothesis: Improvement in PD causes decreasing rates of CSB. The survey methods were assessed using a triaxial accelerometer and self-administered questionnaire. The initial survey involved 96 healthy older persons who participated at a health club of college A in Utazu, Japan (approximate population of 18,450). As previously described [14], we conducted the study from 20 July to 10 September 2016 in the first phase. Since 3 of 96 people canceled and 7 people did not reach the standard measurements of physical activity, we excluded them from analysis. Thus, the remaining 86 respondents were used as reference databases. The second phase involved a similar follow-up survey and was conducted from 20 July to 15 September 2017. Of these respondents, six could not be surveyed. Therefore, we used data based on 80 participants (71.9 ± 5.5 years) [15].

This study was approved by the Shikoku Medical College Ethic Screening Committee (approval number: H27-6), and written informed consent was obtained from each subject.

We received approval from the Shikoku Medical College Ethics Screening Committee (approval number: H28-5), and written informed consent was obtained from each subject.

### 2.2. Clinical Parameters and Measurements

Anthropometric and body composition parameters were evaluated as confounders based on the following parameters: age (years), height (cm), body weight (kg), and body mass index (BMI; kg/m^2^) in 2016 and 2017 [14,15].

### 2.3. Psychological Distress

Data for the K6 scores is cited in our previous papers. Psychological distress was assessed using six items of the Japanese edition of the K6 scale. The K6 is a self-written questionnaire developed by Kessler as a screening test for psychological distress that could effectively discriminate: it is valid and reliable. Subjects answered six items on a 5-point Likert scale, and responses for each item were transformed to scores ranging from 0 to 4 points. The questionnaire consisted of six questions: Over the last month, about how often did you feel: (1) nervous, (2) hopeless, (3) restless or fidgety, (4) so sad that nothing could cheer you up, (5) that everything was an effort, (6) worthless? The subjects were requested to respond by choosing from the following: “all of the time” (4 points), “most of the time” (3 points), “some of the time” (2 points), “a little of the time” (1 point), and “none of the time” (0 points), and the total points was the evaluation level. Thus, the score range was 0–24 [14,15,16,17].

### 2.4. Physical Activity

We recorded using a triaxle accelerometer (Active Style Pro HJA-750C; Omron healthcare company; Kyoto, Japan) for seven consecutive days. Subjects were asked to wear these tools at all times except when it was not possible, such as while swimming and bathing. The standard deviation of the data of 10 s is defined as an average value of acceleration. We adopted seven days including Saturday or Sunday to satisfy wearing 10 h or more per day in this analysis [14,15], where CSB = (time of sedentary behavior continuous for more than 30 min per day) / (awake time in one day; minutes) × 100.

### 2.5. Statistical Analyses

These studies were focused on the elderly with depression but not for those who were healthy. Therefore, it is necessary to clarify the causal relationship between physical and mental factors in healthy elderly people. In this study, “causal relationship” refers to the “Granger” [18] causal relationship. The “Granger causality test” is used to determine the causal relationship by controlling the prior values of each variable and examining the cross-delay effects between them. If the cross-lagged effects of X → Y and Y → X are both significant, a bidirectional causal relationship exists. Furthermore, when only one cross-lagged effect is important, there is no causal relationship between the variables if there is one causal relationship and both cross-delay effects are not significant.

We conducted Structural Equation Modeling (SEM) to clarify the causal relationship between PD and CSB. First, to determine whether the variables from the initial survey could sufficiently predict variables from the second survey, PD and CSB in 2016 were compared with those in 2017. Second, to assess the causal relationship between PD and CSB, we adopted the cross-lagged and synchronous effects models. Finally, to measure the fitness of these models, we used χ^2^ (if *p* > 0.05, it was regarded as conforming to the data), adjusted goodness-of-fit index (AGFI: from 0 to 1, preferably 0.95 or more), comparative fit index (CFI: from 0 to 1, preferably 0.95 or more), and root mean square error of approximation (RMSEA: preferably less than 0.05) [19]. The appropriate sample size for the cross-lagged and synchronous effects models has not yet been established.

SEM is a statistical approach for understanding social phenomena by introducing latent variables that cannot be directly observed and identifying “causal” relationships between the latent variables and observed variables. In this report, we conducted “verifiable causal relationships” using path analysis using only directly observed data.

Also, we conducted a single regression analysis of PD, CSB, and total physical activity (physical activity; METs× h/day) by using amount of change in 2016 and 2017.

All calculations were performed using SPSS version 24 and AMOS version 24 (IBM, Chicago, IL, USA).

## 3. Results

Data are calculated and expressed as mean ± standard deviated (SD) values in Table 1. First, as shown in Figure 1, the K6 scores (PD) and CSB in 2016 were highly correlated in 2017 (PD: β = 0.925, CSB: β = 0.819).

Second, as shown in Figure 1, in the cross-lagged effects model, the path of the model from 2016 K6 scores to 2017 CSB was significant (standardization factor; SF: 0.136, *p* < 0.05); however, the reverse was not (SF: 0.027, *p* = 0.065). Therefore, there is a possibility that the K6 scores in 2016 exerted a causal effect (0.136) on CSB in 2017. Based on the synchronous effects model, in 2017, the path from the K6 scores to CSB was statistically significant (SF: 0.166, *p* < 0.05) while the reverse was not (SF: 0.029, *p* = 0.518) (Figure 2). Therefore, there is a possibility that the K6 scores in 2017 exerted a causal effect (0.166) on CSB in 2017.

Third, to measure the fitness of these models (excluding insignificant paths), the following fitness indexes were obtained and both had high degrees of fitness: χ^2^ = 0.153 (*p* = 0.696), AGFI = 0.990, CFI = 1.000, and RMSEA = 0.000.

Finally, by using amount of change in 2016 and 2017, we showed CSB = 1.145 × K6 + 1.085, adj. *R*^2^ = 0.181, F value = 18.465, *p* < 0.001 (no change after adjustment with other variables). We also showed total physical activity = 0.133 × K6 – 0.095, adj. *R*^2^ = 0.004, F value = 0.262, *p* = 0.262.

## 4. Discussion

Our study had two major findings. PD in 2016 was significantly correlated with PD in 2017. Similarly, CSBs between 2016 and 2017 were also significantly correlated. These results suggest that secular change effects do not have to be considered much.

First, analysis with the cross-lagged effects model showed that the path from K6 in 2016 to CSB in 2017 was still significant even if it was influenced by CSB in 2016. However, the path from CSB in 2016 to K6 in 2017 was not significant. Analysis with the synchronous effects model showed that the path from K6 in 2016 to CSB in 2017 was still significant even if it was influenced by CSB in 2016. However, the path from CSB in 2016 to K6 in 2017 was not significant. These results suggest that K6 in 2016 may have had an impact on CSB in 2017 [20]. Thus, improvement in PD may decrease the ratio of CSB one year later when the cross-lagged and synchronous effects models are used. These results are different from other studies that reported that large amounts of physical activity reduced the likelihood of depression [3,4,7,21]. This discrepancy may be due to the use of cross-sectional studies to clarify the relationship between mental health and physical activity levels. However, the relationship between cause and result has not been sufficiently clarified over an extended period of time in healthy elderly people (K6 scores = 2.4). For example, Kritz-Silverstein D et al. [22] reported that “cross-sectional analyses indicated that before and after adjustment for covariates, exercise was significantly associated with less depressed mood. However, prospective analyses of the 944 persons who attended both clinic visits indicated no association between baseline exercise and either follow-up the Beck Depression Inventory (BDI) score (*p* > 0.10) or change in BDI score between baseline and follow-up (*p* > 0.10). Results confirm that exercisers have less depressed mood”.

Second, in the structural equation model, it was necessary that a nonlogical value did not initially occur in the measurement model and to ascertain that the degree of fitness of the model was good before interpreting the results.

Finally, we showed that CSB of a year ago affected PD of a year after using a single regression analysis. Other total physical activity of a year ago did not affect PD of a year after.

There are some limitations in our study. First, there may be issues with the regression to the mean [23]. Thus, when conducting a longitudinal survey at two points, the high score of the initial survey is higher than the true score of the survey target, and the low score of the survey is lower than the true score of the subject to be surveyed tend. As a result, the high score of the initial survey will fall further by the second survey, but the low score of the initial survey tends to be higher. To alleviate this problem, observations at three or more time points are required. This is because the observed scores randomly fluctuate about the true score, so if the change is measured at more than three time points it will alleviate the problem of regression to the mean [23].

Second, in this study, we did not carefully examine the intervention of the third variable. Even if a causal relationship from x to y is indicated, it may be possible that the relationship is influenced by an unknown third variable. In future studies, a third variable, z, that may influence the estimation of the causal relation between x and y should be considered and incorporated into the model.

Third, since the sample number was less than 100, we used the maximum likelihood method and a simple model [24]; however, in the future, a larger sample number should be obtained in order to use a more complicated model.

Lastly, “psychological well-being was independently associated with attaining and maintaining higher physical activity levels over 11 years” [25]. However, the mechanism by which psychological distress reduces sedentary behavior has not yet been fully elucidated. We need further research to elucidate this mechanism.

For older adults with good psychological well-being, the rate of sedentary behavior (CSB) may not affect PD.

## 5. Conclusions

The results suggest that improvement in PD may affect the ratio of CSB one year later.

## Figures and Tables

**Figure 1 medicina-55-00324-f001:**
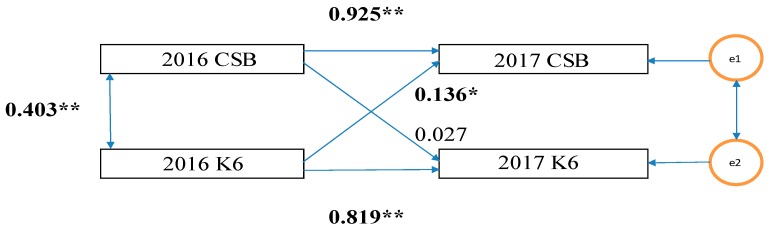
Cross-Lagged Effects Model. CSB, continuous sedentary behavior; K6, K6 scores; and e_1_ and e_2_ error. The numbers in the figure are standardization coefficients. ** *p* < 0.01, * *p* < 0.05.

**Figure 2 medicina-55-00324-f002:**
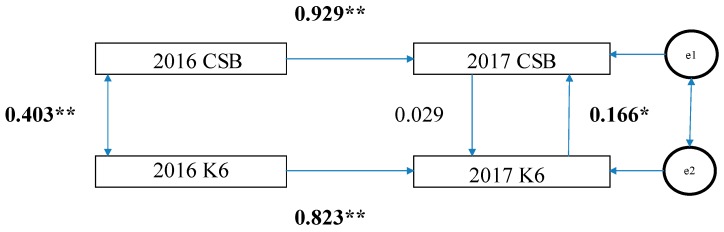
Synchronous Effects Model. CSB, continuous sedentary behavior; K6, K6 scores; and e1 and e2, error. The numbers in the figure are standardization coefficients. ** *p* < 0.01, * *p* < 0.05.

**Table 1 medicina-55-00324-t001:** Clinical characteristics of enrolled subjects.

	2016			2017		
	Mean ± SD	Minimum	Maximum	Mean ± SD	Minimum	Maximum
Number of subjects	80			80		
Age (year)	71.9 ± 5.5	65	85			
Height (cm)	157.2 ± 9.2	138.3	178.4	157.1 ± 9.2	138.3	178.4
Body weight (kg)	56.1 ± 9.8	40.3	86.2	56.0 ± 9.7	40.3	83.1
BMI (kg/m^2^)	22.6 ± 2.8	14.9	29.1	22.6 ± 2.6	14.9	29.1
≤1.5 METs (%/day)	55.1 ± 9.9	35.4	79.9	55.1 ± 10.2	35.4	75.5
CSB: Interrupted of sedentary behavior (%)	14.3 ± 8.6	0.0	40.8	15.4 ± 9.6	0.0	45.4
Total physical activity (METs×h/day)	5.24 ± 2.7	0.5	10.9	5.17 ± 2.2	0.4	9.7
K6 score	2.4 ± 3.0	0	14	2.4 ± 3.1	0	14

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
