# Peer review of "Relationship between Psychological Distress and Continuous Sedentary Behavior in Healthy Older Adults"

_medicina, 2019, doi:10.3390/medicina55070324_

Round 1
Reviewer 1 Report
Thank you for the opportunity to review this manuscript to examine the relationship between psychological distress and prolonged sedentary time at two time points in older adults. The authors found that psychological distress was related to higher levels of prolonged sedentary time one year later. Similarly, psychological distress at the same time point where sedentary time was measured had an influence on higher levels of prolonged sedentary bouts.
Major comments:
-I think the title is a bit misleading with regards to the “causal relationship” between psychological distress and sedentary time in older adults. Given the small sample size of 80 older adults recruited from a health club (likely of high socioeconomic status) and the non-randomized nature of the study, I do not think claims of causality can be made. The term causal may be used as a term when implementing structural equation modeling but for the non-expert reader it will definitely raise a red flag. As such I suggest revising the title.
-Terms physical activity and sedentary time appear to be used interchangeably in the methods section when they are not (PMID: 28599680). In fact, the authors have not considered physical activity as a confounder in their structural equation model. My hypothesis is that physical activity will relate to psychological distress to a greater extent than prolonged sedentary bouts. However, this has not been tested in the present study. Therefore, it is strongly recommended that authors please include physical activity levels from their triaxial accelerometers in their model.
-The discussion section could be significantly improved if the authors would describe in more detail what this study adds to the literature.
Minor comments:
-Where it says in the abstract “For elderly people with good psychological well-being, increase in physical activity may not be very effective in improving mental health.” – It is not made clear in the abstract if physical activity was assessed in the present study, making the conclusion misleading.
-The introduction is a bit disjointed, discussing multiply information. I suggest revising for clarity. For example, discussing the “Granger causality test” would be better suited for the methods section.
-The readability of the manuscript could be improved with a more thorough revision for English grammar and spelling.
-Where it says in the methods, “Finally, to measure the fitness of these models, we used χ2 (if P > 0.05, it was regarded as conforming to the data), adjusted goodness of fit index (AGFI: from 0 to 1, preferably 0.95 or more), comparative fit index (CFI: from 0 to 1, preferably 0.95 or more), and root mean square error of approximation (RMSEA: preferably less than 0.05).” – Can the authors please provide a reference for the ideal level of fit indices?
-For the non-expert readers in structural equation modeling, the authors should provide more details of this approach to examining associations between exposures/outcomes.
-In the discussion, where it says “Our study had three major findings. First, PD in 2016 was significantly correlated with PD in 2017. Similarly, CSB between in 2016 and in 2017 was also significantly correlated. These results suggest that aging is not a significant factor in this study.” I would err on the side of caution here. Measuring sedentary time and psychological distress over one year does not indicate that aging is a factor.
Author Response
Comment 1.
Thank you for the opportunity to review this manuscript to examine the relationship between psychological distress and prolonged sedentary time at two time points in older adults. The authors found that psychological distress was related to higher levels of prolonged sedentary time one year later. Similarly, psychological distress at the same time point where sedentary time was measured had an influence on higher levels of prolonged sedentary bouts.
Major comments:
I think the title is a bit misleading with regards to the “causal relationship” between psychological distress and sedentary time in older adults. Given the small sample size of 80 older adults recruited from a health club (likely of high socioeconomic status) and the non-randomized nature of the study, I do not think claims of causality can be made. The term causal may be used as a term when implementing structural equation modeling but for the non-expert reader it will definitely raise a red flag. As such I suggest revising the title.
(Answer)
We changed the follows according to your instruction.
Title
Causal Relationship between Psychological Distress and Continuous Sedentary Behavior in Healthy Elderly People Older Adults
Lines: 11-13.
Aim to clarify whether improvement in psychological distress (PD) decreases the rate of sedentary behaviors (CSB). Our purpose is to clarify whether psychological distress (PD) affects the rate of sedentary behavior (CSB).
Terms physical activity and sedentary time appear to be used interchangeably in the methods section when they are not (PMID: 28599680). In fact, the authors have not considered physical activity as a confounder in their structural equation model. My hypothesis is that physical activity will relate to psychological distress to a greater extent than prolonged sedentary bouts. However, this has not been tested in the present study. Therefore, it is strongly recommended that authors please include physical activity levels from their triaxial accelerometers in their model.
(Answer)
We added the follows according to your instruction.
Table 1.
Lines: 115-116.
Also, we conducted a single regression analysis of PD, CSB, and total physical activity (physical activity; METs * hours/day) by using amount of change in 2016 and 2017, respectively.
Lines: 133-135.
Finally, by using amount of change in 2016 and 2017, respectively, we showed CSB = 1.145 * K6 + 1.085, adj. R2 = 0.181, F value = 18.465, p < 0.001 (no change after adjustment with other variables). We also showed total physical activity = 0.133 * K6 – 0.095, adj. R2 = 0.004, F value = 0.262, p = 0.262.
Lines: 169-170.
Finally, we showed that CSB of a year ago affected PD of a year after using a single regression analysis. Other, total physical activity of a year ago did not affect PD of a year after.
The discussion section could be significantly improved if the authors would describe in more detail what this study adds to the literature.
(Answer)
We answered the same as the answer to the above question.
Minor comments:
-Where it says in the abstract “For elderly people with good psychological well-being, increase in physical activity may not be very effective in improving mental health.” – It is not made clear in the abstract if physical activity was assessed in the present study, making the conclusion misleading.
(Answer)
We changed the follows according to your instruction.
Lines: 22-24.
For elderly people with good psychological well-being, increase in physical activity may not be very effective in improving mental health.
Lines: 194-195.
5. Conclusions
Improving psychological distress may decrease the rate of CSB one year later in healthy elderly people. The results suggest that improvement in PD may affect the ratio of CSB one year later.
-The introduction is a bit disjointed, discussing multiply information. I suggest revising for clarity. For example, discussing the “Granger causality test” would be better suited for the methods section.
(Answer)
We moved to Methods (lines: 63-101) according to your instruction.
-The readability of the manuscript could be improved with a more thorough revision for English grammar and spelling.
(Answer)
We improved English as much as possible.
-Where it says in the methods, “Finally, to measure the fitness of these models, we used χ2 (if P > 0.05, it was regarded as conforming to the data), adjusted goodness of fit index (AGFI: from 0 to 1, preferably 0.95 or more), comparative fit index (CFI: from 0 to 1, preferably 0.95 or more), and root mean square error of approximation (RMSEA: preferably less than 0.05).” – Can the authors please provide a reference for the ideal level of fit indices?
(Answer)
We added the follows according to your instruction.
Reference [19]
[19] Hooper, D.; Coughlan, J.; Mullen, M. Structural equation modelling: Guidelines for determining model fit. Electronic Journal of Business Research Methods (BJBRM). 2008, 6, 53-60.
-For the non-expert readers in structural equation modeling, the authors should provide more details of this approach to examining associations between exposures/outcomes.
[Answer]
We added the follows according to your instruction.
Lines: 111-114.
SEM is a statistical approach for understanding social phenomena by introducing latent variables that cannot be directly observed and identifying "causal" relationships between the latent variables and observed variables. In this report, we conducted "verifiable causal relationships” using path analysis using only directly observed data.
-In the discussion, where it says “Our study had three major findings. First, PD in 2016 was significantly correlated with PD in 2017. Similarly, CSB between in 2016 and in 2017 was also significantly correlated. These results suggest that aging is not a significant factor in this study.” I would err on the side of caution here. Measuring sedentary time and psychological distress over one year does not indicate that aging is a factor.
[Answer]
We changed the follows according to your instruction.
Lines: 144-148.
Our study had two major findings. First, PD in 2016 was significantly correlated with PD in 2017. Similarly, CSB between in 2016 and in 2017 was also significantly correlated. These results suggest that aging is not a significant factor in this study. Now PD in 2016 was significantly correlated with PD in 2017. Similarly, CSBs between 2016 and 2017 were also significantly correlated. These results suggest that secular change effects do not have to be considered much.

Reviewer 2 Report
This paper aims to study whether psychological distress is associated with sedentary time. This is an interesting question. However, the authors failed to explain their methods. At least in two occasions the phrase "described in our previous papers" was used. This is not effective in communicating what they did. They say that accelerometers were used to assess physical activity, but no further explanation is given, that alone makes this paper not suitable for publication in its current status.
The word "improvement" in psychological distress was used, it suggests a change from baseline to follow-up indicating fewer complains of symptoms, but the analysis indicates that they look at the K6 at baseline and correlate it with future sedentary behavior. Thus, we cannot ascertain whether psychological symptoms improved, worsen or remain unchanged.
Mechanisms that explain how psychological distress can lead to lower time in sedentary behavior need to be explained.
Using additional methods would strengthen the findings and make the message more clear. For example, regression models where sedentary time is used as an outcome and psychological distress scores as exposure, both crude and after adjustment for potential confounders.
Finally, the preferred term is older adults as opposed to elderly
Author Response
Comment 2.
This paper aims to study whether psychological distress is associated with sedentary time. This is an interesting question. However, the authors failed to explain their methods. At least in two occasions the phrase "described in our previous papers" was used. This is not effective in communicating what they did. They say that accelerometers were used to assess physical activity, but no further explanation is given, that alone makes this paper not suitable for publication in its current status.
[Answer]
We added the follows according to your instruction.
Lines: 73-83.
“Psychological distress was assessed using six items of the Japanese edition of the K6 scale. The K6 is a self-written questionnaire developed by Kessler as a screening test for psychological distress that could effectively discriminate: it is valid and reliable. Subjects answered six items on a 5-point Likert scale, and responses for each item were transformed to scores ranging from 0 to 4 points. “The questionnaire consisted of six questions. Over the last month, about how often did you feel: (1) nervous, (2) hopeless, (3) restless or fidgety, (4) so sad that nothing could cheer you up, (5) that everything was an effort, (6) worthless? The subjects were requested to respond by choosing from the following: ‘‘all of the time’’ (4 points), ‘‘most of the time’’ (3 points), ‘‘some of the time’’ (2 points), ‘‘a little of the time’’ (1 point), and ‘‘none of the time’’ (0 point), and the total points was the evaluation level”. Thus, the score range was 0–24.”[14-17].
Lines: 85-90.
We recorded physical activity using a triaxial accelerometer (Active Style Pro HJA-750C, Omron Healthcare, Japan) for 14 consecutive days, as previously described [25]. Subjects were asked to wear the accelerometer at their waist at all times, except when impossible such as during swimming and bathing. The standard deviation in data for 10 seconds was defined as the average value of acceleration. Subjects wore the accelerometer for 10 hours or more each day, in this analysis. Physical activity was evaluated by Σ[metabolic equivalents × h per week (METs•hours/week)], daily step counts (steps/day), daily step hours (hours/day), walking time (minutes/day), and physical activity (≤1.5 METs, 1.6-2.9 METs, 3-5.9 METs) (minutes/day). As the mean of physical activity did not exhibit a normal distribution, we adopted the median. Measurements of physical activity and sedentary behavior were evaluated at baseline and after one year.
The word "improvement" in psychological distress was used, it suggests a change from baseline to follow-up indicating fewer complains of symptoms, but the analysis indicates that they look at the K6 at baseline and correlate it with future sedentary behavior. Thus, we cannot ascertain whether psychological symptoms improved, worsen or remain unchanged.
[Answer]
We added the follows according to your instruction.
Lines: 22-24.
For elderly people with good psychological well-being, increase in physical activity may not be very effective in improving mental health.
Lines: 190-191
For older adults with good psychological well-being, increase in physical activity may not be very effective in improving mental health the rate of sedentary behavior (CSB) may not affect PD.
Lines: 194-195.
5. Conclusions
Improving psychological distress may decrease the rate of CSB one year later in healthy elderly people. The results suggest that improvement in PD may affect the ratio of CSB one year later.
Mechanisms that explain how psychological distress can lead to lower time in sedentary behavior need to be explained.
[Answer]
We added the follows according to your instruction.
Lines: 186-189
Lastly, “psychological well-being was independently associated with attaining and maintaining higher physical activity levels over 11 years.” [25]. However, the mechanism by which psychological distress reduces sedentary behavior has not yet been fully elucidated. We need further research to elucidate this mechanism.
Using additional methods would strengthen the findings and make the message more clear. For example, regression models where sedentary time is used as an outcome and psychological distress scores as exposure, both crude and after adjustment for potential confounders.
(Answer)
We added the follows according to your instruction.
Table 1:
LINES: 115-116.
Also, we conducted a single regression analysis of PD, CSB, and total physical activity (physical activity; METs * hours/day) by using amount of change in 2016 and 2017, respectively.
Lines: 133-135.
Finally, by using amount of change in 2016 and 2017, respectively, we showed CSB = 1.145 * K6 + 1.085, adj. R2 = 0.181, F value = 18.465, p < 0.001 (no change after adjustment with other variables). We also showed total physical activity = 0.133 * K6 – 0.095, adj. R2 = 0.004, F value = 0.262, p = 0.262.
Lines: 169-170.
Finally, we showed that CSB of a year ago affected PD of a year after using a single regression analysis. Other, total physical activity of a year ago did not affect PD of a year after.
Finally, the preferred term is older adults as opposed to elderly
(Answer)
We changed the title according to your instruction.
Title:
Causal Relationship between Psychological Distress and Continuous Sedentary Behavior in Healthy Elderly People Older Adults
Elderly in other places also changed to older adults

Round 2
Reviewer 1 Report
The authors have done a good job addressing the reviewer comments. I accept the manuscript as is with the revisions.
Reviewer 2 Report
None